# Impact of Hormone Replacement Therapy on Risk of Ovarian Cancer in Postmenopausal Women with De Novo Endometriosis or a History of Endometriosis

**DOI:** 10.3390/cancers15061708

**Published:** 2023-03-10

**Authors:** Hee Joong Lee, Banghyun Lee, Hangseok Choi, Taehee Kim, Yejeong Kim, Yong Beom Kim

**Affiliations:** 1Department of Obstetrics & Gynecology, Uijeongbu St. Mary’s Hospital, College of Medicine, The Catholic University of Korea, Uijeongbu-si 11765, Gyeonggi-do, Republic of Korea; 2Department of Obstetrics and Gynecology, Inha University Hospital, College of Medicine, Inha University, 27, Inhang-ro, Sinheung-dong, Jung-gu, Incheon 22332, Republic of Korea; 3Medical Science Research Center, College of Medicine, Korea University, 73, Goryeodae-ro, Seongbuk-gu, Seoul 02841, Republic of Korea; 4Department of Obstetrics and Gynecology, Seoul National University Bundang Hospital, 82, Gumi-ro 173 beon-gil, Bundang-gu, Seongnam-si 13620, Gyeonggi-do, Republic of Korea

**Keywords:** endometriosis, hormone replacement therapy, ovarian cancer, post-menopause

## Abstract

**Simple Summary:**

Endometriosis is a risk factor for ovarian cancer. Meta and pooled analyses have shown hormone replacement therapy (HRT) is significantly associated with an increased risk of ovarian cancer. A combination of estrogen and progesterone is currently recommended to improve menopausal symptoms in women with a history of endometriosis. However, the effect of HRT on the malignant transformation of postmenopausal endometriosis remains unclear. Therefore, this study investigated the impact of HRT on ovarian cancer occurrence in 20,608 postmenopausal women with de novo endometriosis or a history of endometriosis using the nationwide cohort study. With the exception of HRT using estrogen alone, HRT did not increase the risk of ovarian cancer in postmenopausal women with de novo endometriosis or a history of endometriosis. HRT, but not estrogen alone, can be used to improve the quality of life in symptomatic women with postmenopausal endometriosis.

**Abstract:**

The effect of hormone replacement therapy (HRT) on the malignant transformation of postmenopausal endometriosis remains unclear. This study aimed to investigate the impact of HRT on ovarian cancer occurrence in postmenopausal women with de novo endometriosis or a history of endometriosis. A total of 10,304 women that received HRT (the HRT group) and 10,304 that did not (the control group) were selected by 1:1 matching those that met the study criteria. Incidences of ovarian cancer (0.3% in the HRT group and 0.5% in the control group) and cumulative incidence rates of ovarian cancer were similar in the two groups. The overall mean duration of HRT was 1.4 ± 2.2 years, but the duration of HRT in women with ovarian cancer was 2.2 ± 2.9 years. After adjusting for co-variables, receipt of HRT, duration of HRT, combined use of estrogen and progesterone, and tibolone were not found to be risk factors for ovarian cancer. However, the use of estrogen alone was found to be a significant risk factor for ovarian cancer (HR 2.898; 95% CI 1.251–6.715; *p* = 0.013). With the exception of HRT using estrogen alone, HRT did not increase the risk of ovarian cancer in postmenopausal women with a history of endometriosis or de novo endometriosis.

## 1. Introduction

Endometriosis is an estrogen-dependent benign disease considered resolved after menopause [1]. However, endometriosis affects 6–10% of premenopausal women and 2–4% of postmenopausal women [2]. Endometriotic lesions have a risk of around 1% for malignant transformation [2,3,4], and after menopause, they may undergo recurrence or malignant transformation [1,5,6].

Endometriosis is a risk factor for epithelial ovarian cancer (EOC), which accounts for about 90% of ovarian cancer cases, especially clear-cell and endometrioid subtypes [7,8]. Ovarian cancer is the most common cause of death among women with gynecologic cancer [9]. Women with ovarian cancer have a median age at diagnosis of 57–63 years [10], and therefore, many women are diagnosed with ovarian cancer after menopause. Meta and pooled analyses have shown hormone replacement therapy (HRT) is significantly associated with an increased risk of ovarian cancer [11,12]. Furthermore, the risk of ovarian cancer is reported to be higher for current-or-recent users and serous and endometrioid subtypes as well as to increase with HRT duration (especially for durations of ≥10 years) [11,12].

It has been proposed that HRT may reactivate endometriotic lesions and stimulate malignant transformation in women with a history of endometriosis [1,13,14,15,16,17]. However, only a few small studies, namely one randomized controlled trial (RCT) and two retrospective cohort studies, have compared endometriosis recurrence with respect to HRT in postmenopausal women with a history of endometriosis [18,19,20]. Therefore, due to the lack of high-quality evidence, the absolute risks of endometriosis recurrence and malignant transformation cannot be determined, and the impact of HRT use on these risks is unknown [1].

Low-quality evidence suggests that in women with a history of endometriosis, HRT may be used to improve menopausal symptoms [13,21]. However, estrogen alone appears to have a higher risk of endometriosis recurrence and malignant transformation of endometriomas than estrogen and progesterone in combination [1,13,22], and tibolone seems to be safer than estrogen with or without progesterone [23]. Based on available results, combined estrogen and progesterone is currently recommended in women with a history of endometriosis [13,21], though tibolone may be considered [13].

Currently, the effect of HRT on the malignant transformation of recurrent or de novo postmenopausal endometriosis remains unclear. This retrospective, nationwide study was undertaken to determine the impact of HRT on ovarian cancer occurrence in postmenopausal women with a history of endometriosis or de novo endometriosis using Korean Health Insurance Review and Assessment Service (HIRA) data.

## 2. Materials and Methods

South Korea has a universal health coverage system, known as National Health Insurance, that covers ~98% of the Korean population [24]. HIRA claims data include ~23 million women per year [24]. In the present study, we used the claims data of women with diagnostic codes for endometriosis first registered in HIRA between 1 January 2007 and 31 December 2020. The study was approved by the Institutional Review Board of The Catholic Medical Center at the Catholic University of Korea (No. UC21ZESI0126) on 24 September 2021. The requirement for informed consent was waived because the HIRA dataset uses anonymous identification codes to protect personal information as required by the Korean Bioethics and Safety Act.

The codes used to select eligible patients were based on the 10th revision of the International Statistical Classification of Diseases and Related Health Problems (ICD-10), Health Insurance Medical Care Expenses (2017 and 2018 versions), and HIRA Drug Ingredients Codes. Women with endometriosis were defined to have diagnostic codes for endometriosis (ICD-10: N80X) with surgery codes within 60 days before or after an initial diagnostic code. The following exclusion criteria were applied: women ≤49 years old at last clinic visit; women with N80X codes registered between 1 January 2007 and 31 December 2007 (washout period); women diagnosed with ovarian cancer before the date of the first diagnostic code for endometriosis. Women that received HRT (the HRT group) and those that did not (the control group) were identified via 1:1 matching according to age at last clinic visit. Women diagnosed with ovarian cancer before receiving HRT and those diagnosed within one year of HRT receipt were excluded (Figure 1). In addition, women diagnosed with ovarian cancer before menopause or within one year after menopause were excluded from the control group. In this study, in the HRT group, the date of menopause was defined as the date of HRT commencement, whereas in the control group, the date of menopause was defined by matching with the HRT group. Thus, women diagnosed with ovarian cancer in the HRT or control groups before HRT or menopause or within one year after HRT or menopause were excluded (Figure 1).

Women with ovarian cancer were defined to have one or more diagnostic codes for ovarian cancer (ICD-10: C56x) and V193; the V code is a special code for women with any ICD-10 cancer code in South Korea and was established by the Korean Ministry of Health and Welfare in 2008. Women in the HRT group received HRT prescriptions for ≥28 days, whereas women in the control group did not receive a prescription for HRT. Low socioeconomic status (SES) was defined as the use of Medicaid as National Health Insurance. Charlson Comorbidity Indices (CCIs) were calculated using data obtained between 365 days and 1 day before last clinic visit, as described by Quan [25]. Surgery was defined using surgery codes for fulguration, ovarian cystectomy, bilateral or unilateral salpingo-oophorectomy (BSO or USO), or hysterectomy and a concurrent diagnostic code for endometriosis, or surgery codes for fulguration, ovarian cystectomy, BSO, or USO or hysterectomy within 60 days before or after the date of the diagnostic code for endometriosis. Hormone therapy was defined as prescription codes for hormone therapy (combined estrogen and progesterone, estrogen alone, or tibolone).

### Statistical Analyses

The analysis was conducted using R version 3.5.1 (R Foundation for Statistical Computing, Vienna, Austria), and SAS version 9.4 (SAS Institute Inc., Cary, NC, USA) was used to explore and modify big data [26,27]. In the HRT and control groups, women with the same ages at their last clinic visits were considered homogeneous, and thus, the two groups were 1:1 matched for age. The independent t-test was used to analyze continuous variables, and logistic regression analysis, adjusted or not for confounding factors, was used to identify associations between the independent risk factors in each group. Kaplan–Meier curves were constructed using the log-rank test to examine the cumulative incidence rates of ovarian cancer. Trend analysis was conducted using the chi-squared test for trends in proportions, and associations between variables and ovarian cancer were identified via Cox Proportional Hazard Regression analysis with or without adjustment for confounders. Two-tailed tests were used throughout. The mean imputation method was used to account for missing values, and *p* values of <0.05 were considered statistically significant.

## 3. Results

Initially, the data of 115,552 women with a diagnostic code for endometriosis first registered by HIRA between 2007 and 2020 were extracted. Of these, 20,608 women met the study eligibility criteria (Figure 1), and 10,304 (50.0%) were assigned to the HRT group and 10,304 (50.0%) to the control group.

### 3.1. Characteristics of Women with Endometriosis According to Receipt of HRT

The baseline characteristics of the 20,608 study subjects are shown in Table 1. Mean age at last clinic visit in both groups was 55.0 ± 4.6 years, and ages ranged between 50–90 years (Appendix A). Age at endometriosis diagnosis was lower in the HRT group than in the control group (48.9 ± 5.8 vs. 49.6 ± 5.7 years, *p* < 0.001), and ages ranged between 37–89 years (Appendix A). The duration of HRT was 1.4 ± 2.2 years and ranged between 1–11 years (Appendix A). The rate of HRT use before endometriosis diagnosis was 26.0%, and the rate of HRT use after endometriosis diagnosis was 74.0%.

### 3.2. Incidences and Characteristics of Ovarian Cancer According to Receipt of HRT in Women with Endometriosis

The incidences of ovarian cancer were very low (0.3% in the HRT group and 0.5% in the control group). Receipt of HRT was not associated with the occurrence of ovarian cancer (Table 2). Cumulative incidence rates of ovarian cancer were similar in the two groups (Figure 2), and the incidences of ovarian cancer did not change with time in the HRT and control groups (*p* = 0.799 and 0.183, respectively).

Times between endometriosis diagnosis and ovarian cancer diagnosis were similar in the two groups (1.9 ± 2.9 vs. 1.7 ± 2.7 years) (Table 2) and ranged between 1–11 years (Appendix A). Mean time from HRT commencement to ovarian cancer diagnosis was 5.7 ± 3.4 years (Table 2) and ranged between 2 and 11 years (Appendix A). Duration of HRT in women with ovarian cancer was 2.2 ± 2.9 years (Table 2) and ranged between 1 and 11 years (Appendix A). Mean ages at ovarian cancer diagnosis were similar in the HRT and control groups (55.4 ± 8.1 vs. 56.1 ± 7.0 years, respectively) (Table 2) and ranged between 42 and 76 years (Appendix A).

### 3.3. Risk Factors of Ovarian Cancer According to Receipt of HRT in Women with Endometriosis

Multivariate analysis adjusted for potential confounders showed that the risk of ovarian cancer increased significantly with age at endometriosis diagnosis in the control group (HR 1.066; 95% CI 1.022–1.111; *p* = 0.003) but not in the HRT group. In both groups, the risk of ovarian cancer increased significantly with respect to time after endometriosis diagnosis (the HRT group: HR 1.385; 95% CI 1.210–1.585; *p* < 0.001) (the control group: HR 1.216; 95% CI 1.099–1.347; *p* = 0.001) and the number of surgeries for endometriosis (for the HRT group, HR 5.206, 95% CI 1.631–16.612, *p* = 0.005, and for the control group, HR 7.605, 95% CI 3.230–17.905, *p* < 0.001). However, the risk of ovarian cancer was significantly reduced via hysterectomy for disease except ovarian cancer, in both groups (for the HRT group, HR 0.111, 95% CI 0.036–0.344, *p* = 0.001, and for the control group, HR 0.031, 95% CI 0.007–0.137, *p* < 0.001) (Table 3).

Receipt of HRT, HRT duration, receipt of combined estrogen and progesterone, and receipt of tibolone were not found to be risk factors for ovarian cancer. However, the use of estrogen alone was found to be a significant risk factor for ovarian cancer (HR 2.898; 95% CI 1.251–6.715; *p* = 0.013) (Table 3).

## 4. Discussion

In the present study, the incidences of ovarian cancer were very low and similar in postmenopausal women with a history of endometriosis or de novo endometriosis administered or not administered HRT, and cumulative incidence rates of ovarian cancer were similar in these two groups. Moreover, incidences of ovarian cancer did not change with time in either group. Multivariate analysis adjusted for co-variables showed the use of HRT was not a risk factor of ovarian cancer and that the duration of HRT, use of combined estrogen and progesterone, and use of tibolone were not risk factors of ovarian cancer. However, the use of estrogen alone was found to increase the risk of ovarian cancer significantly.

Age at menopause for Korean women is ~50 years [28]. Therefore, women aged ≤49 years at their last clinic visit were excluded from the present study. HRT is usually prescribed to perimenopausal or postmenopausal women to ameliorate menopausal symptoms. In the present study, based on HRT use before or after endometriosis diagnosis, 26.0% of the women with HRT may have had de novo postmenopausal endometriosis and 74.0% have a history of endometriosis.

Endometriosis and HRT are associated with an increased risk of ovarian cancer [7,8,11,12]. Therefore, we expected that women with endometriosis who received HRT would have a higher risk of ovarian cancer than those that did not receive HRT. However, HRT did not increase the risk of ovarian cancer or influence cumulative incidence rates. Moreover, HRT did not change the incidence of ovarian cancer with respect to time. The incidence of ovarian cancer determined in the present study concurs with that found in a previous large-scale cross-sectional study (n = 4331; 0.2%) [29].

In previous meta and pooled analyses, the risk of ovarian cancer was found to increase significantly with HRT duration [11,12], but receipt of estrogen alone for 1 to <10 years in women >50 was not associated with ovarian cancer risk [12]. We found HRT duration was not a risk factor for ovarian cancer in postmenopausal women with a history of endometriosis or de novo endometriosis. We attribute this difference to the short duration of HRT. Furthermore, the duration of HRT was longer in women with ovarian cancer than in all women that underwent HRT (2.2 ± 2.9 vs. 1.4 ± 2.2 years). Furthermore, in our study, women developed ovarian cancer ~3.5 years after stopping HRT.

Previous studies and the ESHRE guideline recommend that in women with a history of endometriosis, combined estrogen and progesterone may be used to treat postmenopausal symptoms or until at least the age of natural menopause after surgical menopause [13,21]. However, estrogen alone is not recommended because of an elevated risk of malignant transformation [13,21]. In a small RCT (n = 21), tibolone resulted in a non-significantly better moderate pelvic pain rate than transdermal estradiol with or without progesterone (9% vs. 40%) after treatment for one year in women with residual pelvic endometriosis [23], and as a result, tibolone was recommended as a safe hormonal treatment for postmenopausal women with a history of endometriosis [13,23]. Likewise, in our study, neither tibolone nor combined estrogen and progesterone were risk factors for ovarian cancer. On the other hand, estrogen alone was identified as a risk factor for ovarian cancer. Thus, the present study shows that unopposed estrogen increases the risk of malignant transformation of endometriotic foci, which supports previous studies and the ESHRE guideline [1,13,21].

In a nationwide cohort study (n = 45,790), the incidence of ovarian cancer increased with time after endometriosis diagnosis (standardized incidence ratios for ovarian cancer were 1.51 at 1–4 years and 1.78 at 5–9 years post-diagnosis) [8]. Interestingly, in our study, women that received HRT underwent more surgeries for endometriosis, and regardless of HRT use, the risk of ovarian cancer increased with the number of surgeries, which suggests the risk increased with time after endometriosis diagnosis.

The CNGOF/HAS clinical practice guidelines mention that concomitant hysterectomy during an endometriotic lesion resection might reduce recurrence rates as compared with the endometriotic lesion resection alone [30]. In the present study, women in the HRT group received ovarian cystectomy, BSO, or USO for endometriosis more frequently than women without HRT but underwent fewer hysterectomies for endometriosis or diseases not including ovarian cancer. However, regardless of HRT use, hysterectomy for disease except ovarian cancer reduced the risk of ovarian cancer, which suggests hysterectomy reduced the risk of endometriosis recurrence.

This nationwide, population-based cohort study is the first to investigate the impact of HRT on the incidence of ovarian cancer in postmenopausal women with a history of endometriosis or de novo endometriosis. Limitations of this study are mainly related to the use of claims data. First, diagnostic and prescription codes were used to define diseases and treatments and medical records were not reviewed, and thus, a small number of women with incorrect codes may have been misclassified. Second, we were not able to determine the times of menopause in either study group because the HIRA dataset did not provide this information. Therefore, we defined the date at menopause in both groups as the date when women in the HRT group first received HRT. Third, we included women first diagnosed with endometriosis from 1 January 2007 using the HIRA dataset. However, it is possible women diagnosed with endometriosis before 2007 and subsequently suffered recurrence were included. Furthermore, we were unable to precisely determine when women were diagnosed with endometriosis.

## 5. Conclusions

This retrospective, nationwide study based on HIRA claims data shows that, excepting estrogen alone, HRT does not increase the risk of ovarian cancer in postmenopausal women with a history of endometriosis or de novo endometriosis. Our results indicate HRT, but not estrogen alone, can be used to improve the quality of life in symptomatic postmenopausal women with a history of endometriosis or de novo endometriosis. We recommend that a large-scale RCT be undertaken to confirm our results.

## Figures and Tables

**Figure 1 cancers-15-01708-f001:**
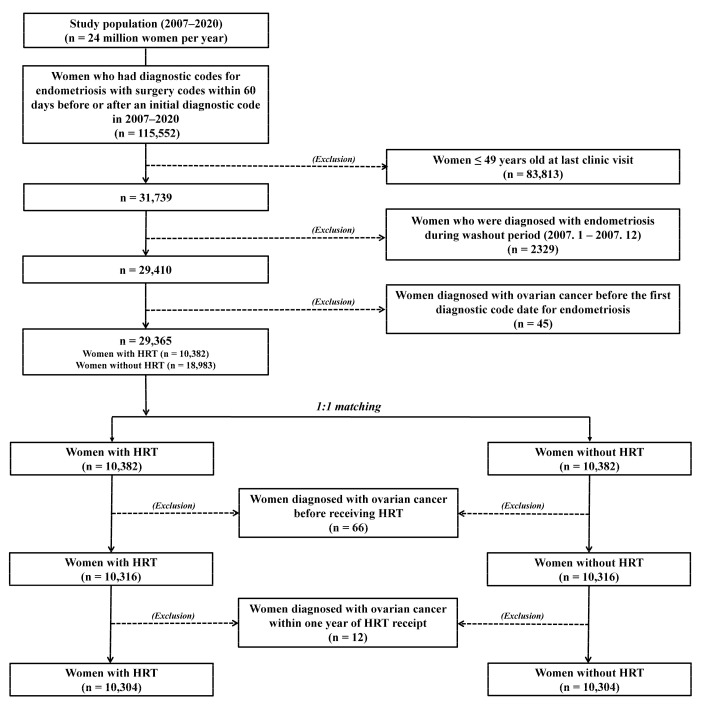
Flow chart showing selection of eligible women.

**Figure 2 cancers-15-01708-f002:**
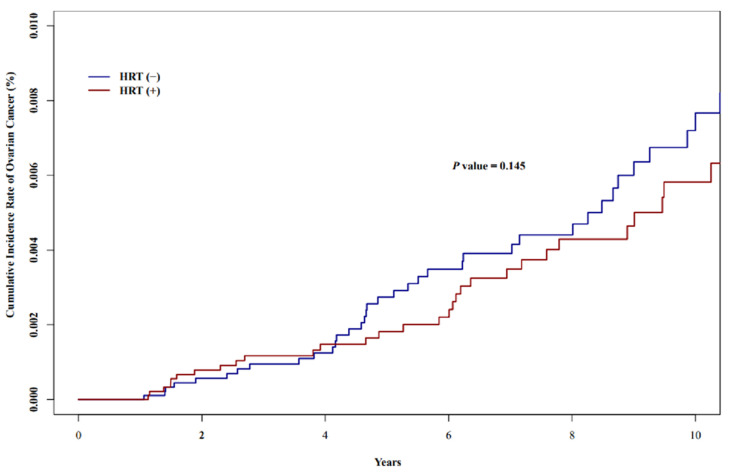
Incidence of ovarian cancer according to HRT in women with endometriosis (HIRA claims data 2008–2020).

**Table 1 cancers-15-01708-t001:** Characteristics according to HRT in women with endometriosis (HIRA claims data 2008–2020).

	Total	HRT (−)	HRT (+)		
	n = 20,608 (100.0%)	n = 10,304 (50.0%)	n = 10,304 (50.0%)	OR (95% CI)	*p* Value ^a^
Age at last clinic visit (years)	55.0 ± 4.6	55.0 ± 4.6	55.0 ± 4.6		
SES at last clinic visit					
Mid- or high-SES	20,078 (97.4)	10,067 (97.7)	10,011 (97.2)	ref	
Low SES	530 (2.6)	237 (2.3)	293 (2.8)	1.243 (1.045–1.479)	0.014
CCI at last clinic visit					
0	11,675 (56.7)	5963 (57.9)	5712 (55.4)	ref	
1	4601 (22.3)	2258 (21.9)	2343 (22.7)	1.083 (1.012–1.160)	0.022
2	2446 (11.9)	1168 (11.3)	1278 (12.4)	1.142 (1.047–1.246)	0.003
3	1028 (5.0)	491 (4.8)	537 (5.2)	1.142 (1.005–1.297)	0.042
Over 4	858 (4.2)	424 (4.1)	434 (4.2)	1.069 (0.930–1.228)	0.349
Age at endometriosis diagnosis (years)	49.3 ± 5.7	49.6 ± 5.7	48.9 ± 5.8	0.978 (0.973–0.982)	<0.001
Year of endometriosis diagnosis					
2008	1670 (8.1)	703 (6.8)	967 (9.4)	ref	
2009	1543 (7.5)	710 (6.9)	833 (8.1)	0.853 (0.742–0.981)	0.025
2010	1576 (7.7)	701 (6.8)	875 (8.5)	0.907 (0.790–1.043)	0.171
2011	1369 (6.6)	630 (6.1)	739 (7.2)	0.853 (0.738–0.985)	0.030
2012	1217 (5.9)	561 (5.4)	656 (6.4)	0.850 (0.733–0.985)	0.030
2013	1442 (7.0)	701 (6.8)	741 (7.2)	0.769 (0.667–0.886)	0.001
2014	1725 (8.4)	835 (8.1)	890 (8.6)	0.775 (0.677–0.887)	0.001
2015	1401 (6.8)	700 (6.8)	701 (6.8)	0.728 (0.631–0.840)	<0.001
2016	1297 (6.3)	674 (6.5)	623 (6.1)	0.672 (0.581–0.778)	<0.001
2017	1931 (9.4)	1050 (10.2)	881 (8.6)	0.610 (0.535–0.696)	<0.001
2018	2244 (10.9)	1230 (11.9)	1014 (9.8)	0.599 (0.527–0.681)	<0.001
2019	1795 (8.7)	991 (9.6)	804 (7.8)	0.590 (0.516–0.675)	<0.001
2020	1398 (6.8)	818 (7.9)	580 (5.6)	0.516 (0.446–0.595)	<0.001
Number of surgeries for endometriosis					
1	20,050 (97.3)	10,088 (97.9)	9962 (96.7)	ref	
2	544 (2.6)	214 (2.1)	330 (3.2)	1.562 (1.312–1.859)	<0.001
Over 3	14 (0.1)	2 (0.0)	12 (0.1)	6.076 (1.360–27.139)	0.018
Methods of surgery for endometriosis					
Fulguration	655 (3.2)	311 (3.0)	344 (3.3)	1.121 (0.959–1.310)	0.150
Ovarian cystectomy	13,302 (64.6)	6536 (63.4)	6766 (65.7)	1.115 (1.058–1.176)	0.001
BSO or USO	2839 (13.8)	1373 (13.3)	1466 (14.2)	1.090 (1.008–1.180)	0.031
Hysterectomy	5329 (25.9)	2919 (28.3)	2410 (23.4)	0.788 (0.741–0.839)	<0.001
Hysterectomy for disease except ovarian cancer	5311 (25.8)	2909 (28.2)	2402 (23.3)	0.773 (0.726–0.823)	<0.001
Duration of HRT (years)			1.4 ± 2.2		
Types of HRT medication					
Combined estrogen and progesterone			3672 (35.6)		
Estrogen alone			4622 (44.9)		
Tibolone			5265 (51.1)		
Time between endometriosis diagnosis and beginning of HRT					
Rate of HRT use before endometriosis diagnosis			2680 (26.0)		
Time of HRT use before endometriosis diagnosis (years)			0.9 ± 1.9		
Time of HRT use after endometriosis diagnosis (years)			1.8 ± 2.1		
Rate of HRT use after endometriosis diagnosis			7624 (74.0)		
Time of HRT use after endometriosis diagnosis (years)			1.3 ± 1.9		

BSO, bilateral salpingo-oophorectomy; CCI, Charlson comorbidity index; HIRA, Health Insurance Review and Assessment Service; HRT, hormone replacement therapy; n, number; OR, odds ratio; SES, socioeconomic status; USO, unilateral salpingo-oophorectomy. All values are expressed as mean ± standard deviation or number (%). ^a^ Univariate logistic regression.

**Table 2 cancers-15-01708-t002:** Incidence and characteristics of ovarian cancer according to HRT in women with endometriosis (HIRA claims data 2008–2020).

	Total n = 20,608 (100.0%)	HRT (−) n = 10,304 (50.0%)	HRT (+) n = 10,304 (50.0%)	OR (95% CI)	*p* Value	Adjusted OR (95% CI) ^a^	*p* Value ^a^
Ovarian cancer							
Negative	20,529 (99.6)	10,258 (99.6)	10,271 (99.7)	ref	0.145	ref	0.283
Positive	79 (0.4)	46 (0.5)	33 (0.3)	0.717 (0.458–1.121)		0.780 (0.495–1.146)	
Time between endometriosis diagnosis and ovarian cancer diagnosis (years)	1.8 ± 2.8	1.7 ± 2.7	1.9 ± 2.9		0.867		
Time between beginning of HRT and ovarian cancer diagnosis (years)			5.7 ± 3.4				
Duration of HRT use in women with ovarian cancer (years)			2.2 ± 2.9				
Age at ovarian cancer diagnosis (years)	55.8 ± 7.4	56.1 ± 7.0	55.4 ± 8.1		0.663		

HIRA, Health Insurance Review and Assessment Service; HRT, hormone replacement therapy; n, number; OR, odds ratio. Values are expressed as means ± standard deviations or numbers (%). ^a^ The data were adjusted for all risk factors (age at endometriosis diagnosis, year of endometriosis diagnosis, number of surgeries for endometriosis, hysterectomy for disease except ovarian cancer).

**Table 3 cancers-15-01708-t003:** Associations between risk factors and ovarian cancer occurrence according to HRT in women with endometriosis (HIRA claims data 2008–2020).

	Total		HRT (−)		HRT (+)	
	HR (95% CI)	*p* Value	HR (95% CI)	*p* Value	HR (95% CI)	*p* Value
Unadjusted HR						
Age at endometriosis diagnosis per 5 years (years)	1.064 (1.037–1.091)	<0.001	1.053 (1.016–1.091)	0.004	1.074 (1.034–1.115)	0.001
Year of endometriosis diagnosis	1.153 (1.086–1.223)	<0.001	1.085 (1.001–1.176)	0.047	1.232 (1.124–1.350)	<0.001
Number of surgeries for endometriosis	3.816 (1.941–7.500)	0.001	6.116 (2.558–14.624)	<0.001	2.599 (0.863–7.829)	0.070
Hysterectomy for disease except ovarian cancer	0.289 (0.126–0.667)	0.004	0.106 (0.026–0.442)	0.002	0.791 (0.277–2.264)	0.662
HRT	0.718 (0.459–1.123)	0.147				
Duration of HRT per year					1.026 (0.917–1.149)	0.651
Types of HRT medication						
Combined estrogen and progesterone					1.044 (0.525–2.077)	0.902
Estrogen alone					2.737 (1.272–5.892)	0.010
Tibolone					0.766 (0.384–1.528)	0.449
Adjusted HR ^a, b^						
Age at endometriosis diagnosis per 5 years (years)	1.056 (1.021–1.091) ^a^	0.001 ^a^	1.066 (1.022–1.111) ^b^	0.003 ^b^	1.025 (0.966–1.087) ^b^	0.414 ^b^
Year of endometriosis diagnosis	1.273 (1.174–1.380) ^a^	<0.001 ^a^	1.216 (1.099–1.347) ^b^	0.001 ^b^	1.385 (1.210–1.585) ^b^	<0.001 ^b^
Number of surgeries for endometriosis	6.375 (3.186–12.757) ^a^	<0.001 ^a^	7.605 (3.230–17.905) ^b^	<0.001 ^b^	5.206 (1.631–16.612) ^b^	0.005 ^b^
Hysterectomy for disease except ovarian cancer	0.064 (0.027–0.153) ^a^	<0.001 ^a^	0.031 (0.007–0.137) ^b^	<0.001 ^b^	0.111 (0.036–0.344) ^b^	0.001 ^b^
HRT	0.937 (0.592–1.483) ^a^	0.782 ^a^				
Duration of HRT per year					0.974 (0.857–1.107) ^b^	0.684 ^b^
Types of HRT medication						
Combined estrogen and progesterone					1.127 (0.539–2.357) ^b^	0.751 ^b^
Estrogen alone					2.898 (1.251–6.715) ^b^	0.013 ^b^
Tibolone					1.057 (0.495–2.257) ^b^	0.886 ^b^

CI, confidence interval; HIRA, Health Insurance Review and Assessment Service; HR, hazard ratio; HRT, hormone replacement therapy. ^a^ The data were adjusted for all risk factors (age at endometriosis diagnosis, year of endometriosis diagnosis, number of surgeries for endometriosis, hysterectomy for disease except ovarian cancer, and HRT). ^b^ The data were adjusted for all risk factors (age at endometriosis diagnosis, year of endometriosis diagnosis, number of surgery for endometriosis, hysterectomy for disease except ovarian cancer, duration of HRT, and types of HRT medication).

## Data Availability

The data that support the findings of this study are available from the Health Insurance Review and Assessment Service (HIRA), but restrictions apply to the availability of these data, which were used under license for the current study and so are not publicly available. Data are, however, available from the authors upon reasonable request and with permission of the HIRA.

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
