# Peer review of "Impact of Hormone Replacement Therapy on Risk of Ovarian Cancer in Postmenopausal Women with De Novo Endometriosis or a History of Endometriosis"

_cancers, 2023, doi:10.3390/cancers15061708_

Round 1

Reviewer 1 Report

Lee et al. submitted their article titled “Risk of Ovarian Cancer posed by hormone replacement therapy in postmenopausal women with de-novo or a history of endometriosis” to Cancers.

The study explores the relationship between hormone replacement therapy (HRT) and ovarian cancer occurrence in postmenopausal women with de-novo or a history of endometriosis. The manuscript is well-written and provides informative insights.

The authors highlight the increased risk of ovarian cancer in women with endometriosis and investigate whether HRT use further increases this risk. However, their findings suggest that HRT, except for estrogen-only therapy, does not increase the risk of ovarian cancer in postmenopausal women with de-novo or a history of endometriosis.

Despite the study’s conclusion, the current title of the manuscript may mislead readers into believing that HRT is a risk factor for ovarian cancer in postmenopausal women. Therefore, I recommend that the authors revise the title to accurately reflect the study’s results to avoid any potential misinterpretation.

Author Response

I attached a file.

Reviewer 2 Report

This is an interesting study indicating that the combined hormone replacement therapy (HRT) is not associated with significant increase of the risk of ovarian cancer in women with endometriosis.

This study is based on the subjects selected from the national database.

To my opinion, the study would become significantly more valuable if all cancer types will be considered, with the emphasis on breast and ovarian cancer. I understand the emphasis on the ovarian cancer, because endometriosis is a risk factor for this malignant disease, but still it is important to see the entire picture.

Simple Summary is not attractive: there are many unnecessary details

( Based on low-quality evidence…”;  “ postmenopausal women with de-25 novo or a history of endometriosis” has been repeated several times).

What is the purpose of statistical comparisons in Table 1?  

Author Response

I attached a file.

Round 2

Reviewer 2 Report

Deserves to be accepted